# Recent Advancements of Polysaccharides to Enhance Quality and Delay Ripening of Fresh Produce: A Review

**DOI:** 10.3390/polym14071341

**Published:** 2022-03-25

**Authors:** Wen Xia Ling Felicia, Kobun Rovina, Md Nasir Nur’Aqilah, Joseph Merillyn Vonnie, Kana Husna Erna, Mailin Misson, Nur Fatihah Abdul Halid

**Affiliations:** 1Faculty of Food Science and Nutrition, University Malaysia Sabah, Kota Kinabalu 88400, Malaysia; felicialingling.97@gmail.com (W.X.L.F.); aqilah98nash@gmail.com (M.N.N.); vonnie.merillyn@gmail.com (J.M.V.); mn1911017t@student.ums.edu.my (K.H.E.); 2Biotechnology Research Institute, University Malaysia Sabah, Kota Kinabalu 88400, Malaysia; mailinmisson@ums.edu.my; 3Borneo Marine Research Institute, University Malaysia Sabah, Kota Kinabalu 88400, Malaysia; fatihahhaliddd@ums.edu.my

**Keywords:** polymers, antioxidant, antimicrobial, shelf life, natural preservative

## Abstract

The freshness of fruits and vegetables plays a significant role in consumers’ decision to purchase a product at the supermarket. Fresh-cut products are the latest trend in fulfilling society’s restless needs, and the food industry is faced with the challenge of maintaining the quality of fresh produce. The food industry is concerned with the natural maturation and degradation of fruits and vegetables, primarily due to enzymatic reactions. It has been demonstrated that polysaccharide coatings effectively preserve the freshness of these products, extending their shelf life depending on the preservation method used. This review informs readers about the different types of polysaccharides and their novel applications as natural food preservatives in the past five years (2018–2022). The key findings summarized the properties of the antimicrobial agent, the molecular mechanism of action, coating methods, and formulation for the preservation approach. Additionally, we discuss the scientific factors influencing polysaccharide processing and preservation efficacy, allowing it to be used in post-harvest management.

## 1. Introduction

Fruits and vegetables are rich in micronutrients, including vitamins C and K, minerals, fiber, and phytochemicals, which are all beneficial to human health [1]. Those foods are considered perishable products because they wilt, shrivel, and degrade over time, reducing their commercial viability and consumer preference. Thus, they have a short shelf life, resulting in significant food waste. Approximately half of all manufactured products are discarded per year [2]. Fruit and vegetable waste is abundant, consisting primarily of peels and other inedible parts of produce, representing approximately 25–30% of total waste [3]. Furthermore, the decay of fruits and vegetables is primarily due to the retailer’s quality standard grading system. Non-compliance with quality standards devalues food regarding its intended use and economic value, increasing food waste.

Both intrinsic and extrinsic factors influence post-harvest quality. Intrinsic factors include genetic variables, harvest maturity stage, susceptibility to physiological illness affected by fungal pathogens, and ethylene gas production. Extrinsic factors include harvesting, packaging, shipping, storage, marketing concerns, technologies, such as irradiation and chemical treatments of fruits and vegetables, and environmental factors, including temperature and relative humidity control [4]. Additionally, Elik et al. [5] stated that faulty and ineffective packaging material will not fully protect fresh produce from damage and may even hasten deterioration. Appropriate packaging materials and advanced packaging technologies significantly apply in preserving the quality of harvested produce.

Food packaging research has resurged, most notably with the appearance of edible coatings on more environmentally friendly commodities, which can be used as a modern and advanced food preservation system to maintain the quality of produce. This significant advancement in food packaging technology reduces food and packaging waste while fostering healthier and safer crops. Over the last five years, there has been much groundbreaking news about edible coatings. Parven et al. [6] and Shah and Hashmi [7] found that polysaccharide-based coatings have the potential to change organoleptic properties in fruits, such as mango, papaya, cherry tomato, and capsicum. Color and firmness, for example, increases shelf life by limiting moisture transfer and gas permeability, inhibits the invasion of a fungal pathogen, and improves nutritional value by reducing ascorbic acid losses in fruits, all of which contribute to a well-balanced diet. 

Nonetheless, the surface polymers’ particular nature properties preclude their vast range of intended applications, necessitating their modification via various techniques. For instance, the presence of hydroxyl groups renders the surfaces hydrophilic, a feature that is not inherent in other polymeric surfaces. Thus, intrinsically nonconforming surfaces must undergo treatment to ease their usage in various industrial applications and overcome their significant limitations. Polymer modification is a frequently utilized technique for imposing desired short- and long-term effects necessary for efficient performance. While surface treatments improve the performance of current materials, they also expand the application range of polymers due to the versatility of bulk material qualities, such as elasticity, tensile strength, and density [8]. 

Edible coatings are a thin layer of edible material applied on the surface of a product during post-harvest processing to help maintain the quality of fruits and vegetables while reducing the use of non-biodegradable packaging materials. The significant edge of edible coatings is that they serve as an additional coating that covers the stomata, lowering transpiration and, as a result, reducing weight loss in commodities [9]. This review informs readers about the different types of polysaccharides and their novel applications as natural food preservatives in the past five years (2018–2022). Polysaccharide-based edible coatings can lengthen shelf life and retard ripening. The Food and Agriculture Organization has designated polysaccharide-based coatings that are considered safe for consumers and environmentally friendly, such as natural gums, as Generally Recognized as Safe [10]. The ability of the coatings to be eaten attests to their safety; however, they must also be acceptable to consumers, retaining the product’s original flavor, texture, and appearance while remaining undetectable on the tongue.

## 2. Types and Properties of Polysaccharides

Polysaccharides are classified as homopolysaccharides or heteropolysaccharides, depending on whether they contain one or more types of repeating units, as shown in Figure 1. The glycosidic linkages that link monomers together to form polymers have a significant effect on the physical properties of polysaccharides [11]. Homopolysaccharides include starch, dextran, pullulan, and cellulose, whereas heteropolysaccharides include chitosan, agar, alginate, and natural gums (xanthan gum, konjac glucomannan, and gellan gum) are frequently used as edible coatings. Polysaccharides have significant potential in advanced fruit and vegetable packaging due to their ability to protect food from deterioration.

### 2.1. Homopolysaccharide

#### 2.1.1. Starch

Starch is highly susceptible to water and has a low water vapor barrier capacity due to its hydrophilic essence. However, a coating made of starch can significantly lengthen the life of fruits, vegetables, and other products. Combining starch-based polymers with other hydrophobic substances to reduce their hygroscopicity shows a cost-effective and versatile strategy for producing innovative materials with better properties [12]. Previously, Oyom et al. [13] modified sweet potato starch and fabricated cumin essential oil to develop an edible coating. The modified edible coating displayed positive antifungal impacts on ‘early crisp’ by significantly reducing rot lesion infections on pears, which is commonly caused by *Alternaria alternata*. Aside from that, the developed edible coating has successfully shown to delay changes in respiration rate, weight loss, chlorophyll degradation, color, and firmness. Furthermore, Trinh et al. [14] stabilized cornstarch with beeswax emulsion and cellulose nanocrystals (CNC) to produce an edible coating that significantly reduces deterioration of fruits and other produce. The effect of beeswax and CNC emulsion displayed beneficial impacts on fresh-cut apples, strawberries, and bananas in preserving color and freshness of the products, as well as limiting oxygen activity and minimizing moisture loss. 

On the other hand, Rather et al. [15] used lotus stem starch incorporated with gelatin waste from poultry to prolong shelf-life of cherry tomatoes by retaining its firmness and pH to a greater extent. Besides, the coating material of starch incorporated with gelatin revealed a positive contribution in minimizing weight loss during the storage period. Similarly, Ghoshal and Chopra [16] enhanced the tamarind seed starch with gelatin and incorporated apricot essential oil to improve the physicochemical properties of grapes by reducing weight loss and firmness as well as providing antimicrobial properties. Chen et al. [17] proposed a bilayer film composed of a hydrophobic outer layer and absorbent inner layer of corn starch and polylactic acid. The film is further enriched with eucalyptus essential oil microcapsules to improve the tensile strength, elongation at break, and barrier properties of the film. The enriched starch-based film effectively inhibited the respiration rate of mushroom (*Agaricus bisporus*), reduced consumption of organic matter, and retained its moisture content along with prolonging the shelf life of *A. bisporus* to a greater extent.

Furthermore, Kawhena et al. [18] studied the effect of the combination of starch and gum arabic coating enhanced with two polyliners on pomegranate fruit. Coated pomegranate fruit showed reduction in weight loss and respiration rate. Additionally, the coating retained the total soluble solids while preventing decay on the appearance of the fruit. Besides, Kusnadi et al. [19] employed maize starch in the development of edible coatings along with k-carrageenan and agar as matrix. The developed coating has been reported to retain color, delay browning, and reduce mass shrinkage of fresh cut apples (*L. plantarum*). It was also reported that the proposed film was actively against the growth of *Escherichia coli*. Starch-based coatings commonly show low water barrier capacity; therefore, enrichment with other hydrophobic compounds, such as essential oils, can help in improving the water vapor barrier properties of the coating. 

#### 2.1.2. Dextran

Dextran is an exopolysaccharide, a flavorless food-grade chemical that is highly biocompatible and non-toxic, and may be used in novel food packaging. Dextran was hypothesized to be an oxygen scavenger and a moisture-resistant compound that acts as an adequate water vapor barrier [20]. In comparison to other polysaccharides, it is an excellent biopolymer for edible coatings that support lengthening the shelf life of food products. Dextran-based coatings on fruits and vegetables have received less attention in the past five years (2017–2022) in the research field. Davidovic et al. [21] optimized dextran-based coating with sorbitol, and they were observed under response surface methodology. The research reported that the coating displayed a favorable impact in reducing water vapor permeability whilst having good tensile strength and elasticity. Davidovic et al. [22] developed an edible coating based on dextran that was synthesized from lactic acid bacteria of *Leuconostoc mesenteroides* and plasticized with polyglycerol. The coating was tested on blueberries to evaluate the quality of the coated fruits. Coated blueberries presented low water vapor permeability and indicated that the film was able to inhibit the fruits from drying out, therefore proving able to efficiently prolong the shelf-life of blueberries than when untreated, as evidenced by lower weight loss, total sugar solids values, and titratable acidity. 

#### 2.1.3. Pullulan

Pullulan is a colorless, odorless, oxygen and carbon dioxide impervious exopolysaccharide derived from the fungus *Aureobasidium pullulans* [23]. Pullulan-based coatings outperform other polymer compounds in adhesion, mechanical strength, and food component resistance [24]. These beneficial properties identify pullulan as edible “active” coatings. Pullulan, a polysaccharide Generally Recognized as Safe (GRAS), was employed in the study to extend fruit shelf life. Antecedently, Ganduri [25] formulated pullulan with calcium chloride and lemon juice to be applied on Rastali and Chakkarakeli bananas. This research demonstrated that coated bananas were well preserved, with the evidence of minimum change in weight, color, browning index, and total and residual sugar. Aside from that, the coating has shown to decrease peelpulp ratio, vitamin c contain, and maintain the firmness of fruit. Meanwhile, Kumar et al. [26] functionalized pullulan/chitosan composite coating with pomegranate peel extract to preserve mango fruit (*Mangifera indica*) with its exceptional antioxidative properties. The enhanced composite film effectively decreased physiological loss in weight, and retained pH, acidity, and total soluble solids of coated mango fruits. Incorporation of pomegranate peel extract into the coating matrix sustained fruit firmness, color, texture, taste, freshness, phenolic content, and antioxidant activity of fruits. Similarly, Kumar et al. [27], Kumar et al. [28], and Kumar et al. [29] exploited pullulan/chitosan/pomegranate peel extract edible coatings to evaluate the effects on quality, sensory characteristics, and shelf-life of litchi fruit (*Litchi Chinensis Sonn.*), green bell peppers, and tomatos (*Solanum lycopersicum* L.), respectively. Indistinguishable findings have been reported for this research; coated fruits exhibited low weight loss, acidity, and color browning, as well as contributed to antioxidant activity. The combined effects of the coating were capable of extending the shelf life of tomatoes by 9 days at temperatures of 23 °C and 4 °C.

Zhou et al. [30] proposed an active edible film based on pullulan and carboxymethyl chitosan fabricated with galangal essential oil to characterize the physical and structural properties and preservation effect on mangoes. The eco-friendly packaging presented a promising alternative in preserving mangoes by inhibiting weight loss and sustaining firmness, titratable acidity, and soluble solids. Most of the pullulan-based coating is commonly synthesized with chitosan polysaccharide due to its antimicrobial properties. Research has been performed to investigate the synergistic effect of pullulan and chitosan on fresh papayas. In general, synergistic interaction of both polysaccharides helps in maintaining post-harvest quality (weight, firmness, color, pH, titratable acidity, vitamin c and soluble solids contents) and lengthening shelf life of fresh papaya [31]. Furthermore, Krasniewska et al. [24] constructed an edible coating solely based on pullulan to improve the quality and shelf life of the highbush blueberry. The study indicated that pullulan coating is capable of protecting perishable fruits by functioning as a barrier against respiration, quality deterioration, and microbial growth. Moreover, Pobiega et al. [32] reduced cherry tomato microbial contamination with an edible coating of pullulan and ethanolic extract of propolis (EEP). The combined effect of the coating has proven to have antimicrobial properties against various foodborne pathogens. Besides, incorporation of EEP enhanced the pullulan action and hence decreased the weight loss and ripening time of cherry tomatoes. 

#### 2.1.4. Cellulose

Cellulose is a polymeric composite that is edible, biodegradable, and may increase nutritional value while reducing synthetic packaging and waste in the environment. It can readily encapsulate antibacterial and antioxidant chemicals [33]. The high concentration of intramolecular hydrogen bonds in cellulose contributes to its water resistance and crystalline structure, making it an excellent coating material due to its flexibility, durability, transparency, and resistance to fats and oils [34]. Additionally, it assists in preserving structural integrity through improved mechanical handling, preventing chlorophyll loss, and maintaining the color of food. As a result, cellulose prevents and reduces microbial degradation during long-term food storage [35].

Carboxymethyl cellulose (CMC) is one of cellulose derivatives. Recently, Yu et al. [36] sustained postharvest quality of pakchoi (*Brassica chinensis* L.) vegetables using CMC-based coating crosslinked with liquid paraffin. The synergistic interaction of ionic bonding of CMC and paraffin significantly reduced moisture loss and enhanced nutrition and appearance of fresh pakchoi. The treated samples displayed low lipid peroxidation due to the scavenging ability of the coating to scavenge reactive oxygen species. Kowalczyk et al. [37] evaluated the physiological, quality, and microbiological count of Brussels sprouts with CMC/candelilla wax emulsion. In contrast, the treated sprout is unaffected in terms of weight loss, texture, and moisture content. Sequentially, the coating decreased rate of respiration hastened chlorophyll degradation and accelerated polyphenol oxidase, and hence reduced sensory attributes of the vegetables. The coating negatively impacts the sample by favoring fungal growth due to the accumulation of ethylene and poor performance of gas exchange. Chen et al. [38] designed a novel edible coating based on CMC incorporated with ethanol extract of *Impatiens balsamina* L. stems, citric acid, sucrose ester, calcium propionate, and glycerol to be applied on “Xinyu” tangerines. The coating displayed both inhibitory effects on mold growth and antioxidant properties on fruits. 

Formerly, Zhang et al. [39] successfully inhibited natural spoilage microorganisms on cantaloupe rind and fresh-cut pulp by creating a surface barrier protection against oxygen-dependent microorganisms. The surface on fruits were coated with cellulose nanofiber-based/chitosan coating enriched with trans-cinnamaldehyde. Saba and Amini [40] functionalized nano-zinc oxide onto cellulose-based active coating to subside mesophilic bacteria count and growth on the arils surface of pomegranate fruits. The cumulative effect of the coating decreased weight loss and maintained overall quality of the fruits. Xie et al. [41] stabilized beeswax onto the cellulose nanofibrils/carboxymethyl chitosan coating matrix. The emulsified coating solution presented good stability and antimicrobial properties by inhibiting *S. aureus* and *E. coli*. Moreover, the coating solution has optimistic potential in keeping berry fruits fresh via its good mechanical and barrier properties. Meanwhile, Liu et al. [42] loaded asparagus waste extract in hydroxyethyl cellulose/sodium alginate edible coating and applied it to the surface of fresh strawberries. The edible coating exhibited inhibition mechanisms against *Penicillium italicum* and, remarkably, sustained the quality of the strawberry. 

### 2.2. Heteropolysaccharide

#### 2.2.1. Chitosan

Chitosan is a polysaccharide that is water-soluble at an acidic pH, making it suitable for use as a coating agent on fresh fruits and vegetables, and has been shown to inhibit pathogenic microorganisms and extend shelf life [43]. Its reactive groups will limit the growth of microorganisms by enhancing the quality, safety, and use of fruits and vegetables. Recently, Shebis et al. [44] synthesized chitosan-based coating enriched with quercetin to enhance the antimicrobial and antioxidative properties of the film. The efficacy of the film was tested on fresh-cut fruit, such as ‘Galia’ melons and ‘Gala’ apples, and successfully reduced microbial spoilage and browning of the fruits. Besides, enhancement with quercetin ameliorates the activity of chitosan by preventing moisture and weight loss. Meanwhile, Liu et al. [45] proposed a composite film composed of carboxymethyl chitosan enriched with Camellia essential oil to prolong the shelf-life of blueberry by improving the physicochemical properties of fruits, such as retaining firmness, decreasing weight loss, and reducing soluble solids formation. Guo et al. [46] loaded chitosan with thymol to retard chestnut decay during storage. The study beneficially influences the starch and nutrient content in soluble sugar along with reducing the respiration rate and weight loss as well as inhibiting the growth of yeast and mold. Correspondingly, Saki et al. [47] combined chitosan coating with thymol to retain quality and prolong the shelf life of fresh fig fruits. The combined coating of chitosan-thymol provided better efficacy in preserving the fruit as compared to solely chitosan coating. Coated fresh fig fruits have low weight loss, respiration rate, TSS, TSS/TA, and fungal decay incidence. Moreover, they show excellent appearance in terms of firmness and color after 20 days of storage at 6 °C. 

Comparably, Qiao et al. [48] investigated the impact of chitosan/nano-titanium dioxide/thymol/tween on cantaloupe fruit quality. The composite film has proven to lengthen shelf life and decrease malondialdehyde content, water activity, polyphenol oxidase, and total soluble solids of the fruits. Zheng et al. [49] constructed an edible film based on a combination of chitosan and acorn starch enriched with eugenol to improve physicochemical, structural, barrier, antioxidant, and antimicrobial properties of the film. Incorporation of eugenol onto the modified film is attested to enhance the film flexibility, hydrophobicity, barrier, antioxidative and antimicrobial properties. Aside from being supplemented with edible essential oils, chitosan-based coating can also be reinforced with nanoparticles to improve the performance, efficacy, and efficiency of the coating. Research done by La et al. [50] investigated the effect of fabricating zinc oxide nanoparticles on chitosan/gum arabic edible coating on the preservation of bananas. The coating considerably increased the quality and shelf life of bananas, including fruit firmness, weight loss, titratable acidity, and reducing sugar. Coated bananas were able to keep their freshness for almost 17 days compared to 13 days for the uncoated bananas. Additionally, the coating was able to provide antimicrobial properties, as the fabricated ZnO nanoparticles on the film matrix are proven to have excellent antibacterial properties against *Staphylococcus aureus*, *Bacillus subtilis*, and *Escherichia coli*. 

Little research has been done to investigate the chitosan/vanillin coating effect on the tomato fruit. The combined edible coating remarkably decreased disease incidence and disease severity by 74.16% and 79%, respectively. Aside from that, the enhanced coating successfully proved to reduce weight loss and retain firmness, soluble solid concentration, and color, as well as reduce respiration and ethylene production rate. The chitosan/vanillin coating also notably reduced the defense enzyme activities, such as peroxidase, polyphenol oxidase, and phenylalanine ammonia-lyase, and increased the lifespan of tomatoes to 25 days at room temperature. Additionally, the combined coating displayed antimicrobial properties by inhibiting *Fusarium oxysporum* fruit rot on tomatoes [51]. Moreover, active chitosan/vanillin coating containing zeolite that exhibits the highest capacity of ethylene scavenger activity delayed the disease incidence of Nam Dok Mai mango fruit by providing the lowest onset of anthracnose disease. As mentioned in previous research, chitosan/vanillin coating was able to retain the physicochemical qualities of mango fruits, including weight loss, firmness, titratable acidity, total soluble solid, and color [52]. Chitosan-based coating supplemented with vanillin and geraniol exhibited bactericidal effects on apple cubes stored under refrigerated temperatures by retarding the growth of mesophiles, psychrotrophs, and yeasts and molds [53]. 

#### 2.2.2. Agar and Alginate

Agar and alginate are hydrophilic polysaccharide coatings made from seaweed of linear copolymers. 3-O-substituted-d-galactopyranosyl units are connected to 3,6-anhydro—d-galactopyranosyl units via (1 → 4) linkages to form agar. Alginate can also produce transparent, homogeneous, and water-soluble coatings but are less permeable to fats, oils, and oxygen [54]. These coating films protect the quality of fresh produce by decreasing moisture and weight loss due to its low water vapor permeability. Furthermore, incorporation of agar and alginate also increased the tensile strength, surface hydrophobicity, and thermal stability of the films through physical entanglement [55]. Post-harvest films and coatings help keep the quality of fruits, such as sweet cherry, peach, and tomato, from shrinkage, oxidative rancidity, oil absorption, flavor, and color loss. Alginate extends fruit and vegetable shelf life by preventing degradation by hampering microbial growth and respiration [56]. Previously, Medina-Jaramillo et al. [57] used alginate-based coating enriched with carvacrol essential oil to develop an edible coating to be applied on Andean blueberries. Coated blueberries outlasted untreated blueberries over the course of three weeks as a result of lower respiration rate and water loss. The coatings also improved the fruits’ appearance and inhibited growth of mesophilic aerobic bacteria and mold/yeast on fruits. Das et al. [58] combined alginate with vanillin to eradicate food pathogens and prolong the lifespan of leafy vegetable lettuce. Coated lettuces were found to significantly reduce weight loss and bacterial loads by efficiently eliminating sessile cells of biofilm-associated foodborne pathogens. Sangsuwan and Sutthasupa [59] encapsulated clove and lavender essential oils along with vanillin onto the alginate/chitosan beads matrix to prevent *Botrytis cinerea* in table grapes. Grapes packed with enriched beads have shown to have no visible sign of mold on grapes until Day 28, while retaining freshness, odor, flavor, firmness and overall acceptability of the grapes compared to the unpacked grapes.

Hu et al. [60] incorporated thyme oil into an alginate-coating matrix to investigate the microbial effect on fresh-cut “Red Fuji” apples. The combined usage of the coating effectively prevents the growth of microorganisms, and reduces weight loss, browning, and respiration rate, as well as retains the firmness of fresh-cut apples. Bambace et al. [61] combined alginate and vanillin to synthesize an edible coating. The synthesized coating was applied on apple cubes and the fruit was treated with high-pressure processing (HPP) in order to increase the performance of the film in retarding microbial growth. Utilization of HPP treatment onto the coating left *L. monocytogenes* and *E. coli* counts below the detection limit while maintaining the firmness and appearance of the apple as well as retaining the phenolic compounds in apple cubes. Additionally, Shakerardekani et al. [62] also enhanced sodium alginate coating with thyme essential oil to investigate its effect on fresh pistachio. Similar findings were reported for the effectiveness of the coating in reducing weight loss and microbial growth while maintaining firmness, color, soluble sugar, and chlorophyll content. 

Instead of solely using alginate-based coating, cocoa was incorporated into the coating matrix to improve the physicochemical, microbiological, and sensory properties of fresh-cut oranges. Coated orange samples retained their highest post-harvest quality over the storage period, including maintaining excellent sensory properties and having low yeast and mesophilic aerobic bacteria count [63]. Sodium alginate-based coating incorporated with thymol has proven to preserve the quality of fresh-cut apples. The prominent antibacterial and antioxidant activities of thymol overcame the non-antibacterial and poor mechanical properties of alginate. Enhancement with thymol essential oils remarkably inhibited foodborne pathogens on fresh-cut apples, such as *Listeria monocytogenes*, *Staphylococcus aureus*, *Salmonella Typhimurium*, and *Escherichia coli* by damaging and inactivating the cell membrane of the pathogens. Aside from that, a combination of thymol/sodium alginate composite coating has shown to have better tensile strength, elongation at break, and capabilities of blocking UV-vis light. The coating also notably scavenged 1,1-diphenyl-2-picrylhydrazyl radicals and considerably decreased weight loss and retained nutrition and color of the apples [64,65]. Incorporation or fabrication of other compounds, such as essential oil onto alginate-based coatings, is essential to improve the efficacy of the coating as standard sodium alginate commonly displays poor water vapor permeability, water solubility, and swelling ratio. 

#### 2.2.3. Mucilage

Mucilage is a hydrophilic hydrocolloid that improves barrier properties in low relative humidity, form slimy abundances that take longer to degrade than natural gums, and form durable edible coatings. Due to their low cost and eco-friendliness, basil, quince, flaxseed, and wild sage are the most frequently used seeds in mucilage-based biodegradable films [66]. Mucilage used in edible films and coatings shows antibacterial and antioxidant properties, making it an ideal platform to improve the appearance of food products [67]. 

Recently, Shahbazi et al. [68] developed an edible coating by combining okra mucilage-quince seed mucilage with bacterial cellulose nanofibers and *Eryngium planum* extract. The proposed coating auspiciously prolonged the shelf-life of fresh strawberries for 12 days under refrigerated temperature. Coated strawberries presented excellent physicochemical (pH, color, weight loss, and titratable acidity), microbial count (yeast and molds count, psychrotrophic bacterial count, and total viable count) and barrier (water vapor resistance, oxygen permeability, and thickness) properties. Nourozi and Sayyari [69] proposed the combined usage of aloe vera gel and basil seed mucilage on preserving the qualitative parameters of apricot fruits. Combination of the polymers notably decreased weight loss, soluble solid content, respiration, and ethylene production rate, hence further reducing the ripening index of apricot fruits. Uncoated fruits displayed fruit softness and low titratable acidity. Combined effects provided better antioxidative activity, thus enhancing the total phenolic content (TPC) and ascorbic acid of the fruits. The utilization of the coating on the fruits demonstrated no changes in organoleptic properties of apricot fruits.

Mohammed et al. [70] also designed an edible coating based on aloe vera gel combined with linseed mucilage to lengthen the lifespan of plums. Indistinguishable from the previous study, the combined effect of aloe vera and mucilage reduces respiration rate, weight loss, pH, titratable acidity, and chroma index. However, this research reported that firmness and soluble solid content of fruits were not influenced by the coating. Kozlu and Elmaci [71] developed an edible coating solely based on quince seed mucilage for the purpose of extending the shelf life of mandarin fruit. This research has proven that mucilage is able to maintain firmness, color, antioxidant activity, and total phenolic content, reduce weight loss, as well as preserve sensory attributes of fruits. Liguori et al. [72] evaluated the performance of the combination of *Opuntia ficus-indica* mucilage and ascorbic acid edible coating on postharvest quality, sensory characteristics, and microbiological counts of strawberry fruit. Uncoated strawberries were negatively influenced as they showed a linear increase in loss in terms of weight, ascorbic acid, and total soluble solid content. However, the proposed coating does not exhibit antimicrobial effects on the fruits. Despite lacking antimicrobial properties, the coating was reported to significantly limit the development of pathogens on the coated strawberry fruit. 

#### 2.2.4. Natural Gums

A recent study critically looked into the use of edible coatings to improve the shelf-life of fresh fruits and vegetables. Unlike manufactured polymers, these natural polymeric polysaccharides are biodegradable, nontoxic, cheap, and readily available. Natural polymers found in plant gums enable the formation of films and coatings with excellent barrier qualities against the transfer of gases, such as moisture, oxygen, and carbon dioxide. Besides, it is linked topically by dipping or spraying onto fruit or vegetable surfaces, then dried by air [73]. Formerly, Hashemi and Jafarpour [74] incorporated *Lactobacillus plantarum* strains onto Konjac-based edible coating to inhibit fungi growth and maintain physicochemical attributes of fresh-cut kiwis. The presence of *L. plantarum* as probiotics considerably decreased the amount of decay and color changes while maintaining chlorophyll and ascorbic content of the fruits. Probiotics treatment notably decreased mold and yeast counts. Meanwhile, Sati and Qubbaj [75] synthesized gum arabic/cactus mucilage edible coating incorporated with calcium chloride to compare the postharvest attributes of coated and uncoated tomato fruits. Combined usage of the coating beneficially reduced the weight loss and decay incidence and retained firmness, color, titratable acidity, and total soluble solids content of the tomato fruits. Criado et al. [76] developed a coating based on gellan gum and enhanced with cellulose nanocrystals to create surface protection and lengthen the shelf life of *Agaricus bisporus* mushrooms. Implementation of nanomaterials exhibited outstanding capabilities in improving the barrier capacity of materials. Results demonstrated that coated mushrooms showed a reduction in color change and respiration rate. Ergin et al. [77] extracted gum exudates from cherry and apricot trees to be used as edible coating material. The extracted gum exudates were applied on strawberry (*Fragaria ananassa*) and loquat (*Eriobotrya japonica*) fruits to improve the antioxidant capacity and phenolic content of the fruits. Additionally, the coating was reported to be heat-resistant up to 400 °C and used as a preserving film in reserving shelf life, organoleptic, and microbiological properties of fruits. 

In addition, Nasiri et al. [78] exploited tragacanth gum along with *Satureja khuzistanica* essential oil to serve as a natural preservative for button mushrooms. This coating lengthened the shelf-life of button mushrooms by maintaining tissue firmness and sensory attributes of the mushrooms as well as reducing microbial count, such as yeast and molds and *Pseudomonas*. Besides, the coating also reduced the rate of decomposition of functional compounds of mushrooms. Sarpong et al. [79] prepared a film based on karaya gum, xanthan gum, and acacia Senegal, and applied it on applied slices. The ameliorative effect of the gum coating inhibited polyphenol oxidase, peroxidase, and ascorbic acid oxidase activities as well as reduced browning indexes. Meanwhile, Saleem et al. [80] implemented usage of gum arabic as edible coatings in prolonging shelf life of persimmon fruits stored under ambient temperature. Treated samples displayed lower weight loss, H_2_O_2_, malondialdehyde content, and membrane leakage. Gum arabic is proven to inhibit enzyme activity, such as polygalacturonase, cellulase, and pectin methylesterase, while, contradictorily, it supports catalase, ascorbate peroxidase, peroxidase, and superoxide dismutase activities. Treated persimmons depicted higher total phenolics, ascorbic acid, antioxidant activity, and titratable acidity.

## 3. The Coating-Forming Agents Designed to Preserve Fruits and Vegetables

Key quality aspects of edible films or coatings are the protective characteristics against water vapor, gases, and chemical migration; physical and mechanical protection; and influence on product appearance, including color and gloss [81]. Post-harvest quality loss of fresh goods is connected to biochemical and physiological changes in the living tissue caused by mass transfer phenomena, such as moisture and oxygen exchanges, undesirable odor absorption, ethylene production, or flavor loss [82]. Post-harvest losses are related to microbial changes, fungal deterioration, increased risk of food-borne disease, and shorter storability. Antibacterial chemicals in the polymeric matrix are another prominent topic in coating design. Besides, the effectiveness of edible coating relies on the wettability of the product and is impacted by both the surface qualities of fresh food and the chemical composition and structure of coating-forming polymers. Plasticizers, surfactants, antimicrobials, and antioxidants can alter the coating process’ efficacy and affect the film thickness during coating development [83]. 

Other factors, such as tensile strength, affect edible coating efficiency. Tensile strength and stretchability are crucial film parameters that can be utilized to monitor the film’s integrity and resistance to environmental stress throughout the coating treatments [84]. Mechanical resistance prevents coating film cracking and protects fruits and vegetables from impact, pressure, and vibrations while in storage [85]. The problems in terms of mass transfer qualities of coatings include reducing water vapor permeability levels to avoid moisture loss, weight loss, or changes in texture, taste, and appearance [86]. Water loss and gain are often seen negatively. The coating must also have low oxygen permeability since this respiration process boosts sugar and other chemical consumption, increasing ethylene production and triggering senescence [82]. A varied minimum oxygen transfer rate may be necessary to avoid unwanted metabolic differences based on the fruit or vegetable’s respiration rate. Coating extensibility and adhesion are essential factors in enhancing coating functions and the quality and appearance of coated fruits and vegetables. Some active chemicals incorporated in the edible coatings may modify the organoleptic character of the coated product, causing unpleasant scents or functional modifications. Active compounds, such as essential oils, can be harmful to plant cells or lose potency when exposed to environmental or dietary elements [87]. Figure 2 shows the direct correlations between coating qualities and the quality variables maintained in fruits and vegetables. The interfacial interaction between the coating-forming chemicals and the surface energy of a product affects how well a coating protects against abrasion.

## 4. Applications and Effects of Polysaccharides Coating on Fruits and Vegetable Postharvest

Fresh food degrades rapidly during harvesting, processing, transit, and storage, with over half of the items ripening and decomposing during these activities. Coatings manufactured from edible materials are essential in this circumstance because they may be applied to fruits and vegetables after harvesting, employing electrospraying, conventional spraying, spreading, dipping, brushing, and layer by layer deposition processes [88]. Polysaccharides coatings are selected for industrial application based on the hydrophobicity, the roughness of the fresh produce surface, and the physical features of each coating properties of the edible coatings, including coating emulsion stability, surface tension, viscosity, cost, density, and drying conditions [89]. Polysaccharide coatings are an innovative method for increasing food quality for customer satisfaction. Non-toxic polysaccharide coatings protect the nutrient content of fresh food while suppressing microbial development, creating a gas and water vapor barrier, and preventing oxidation [90]. Combining active compounds with the polymer matrix improves food’s sensory and nutritional quality. Table 1 displays examples of polysaccharide coatings employed to enhance the quality of various fruits and vegetables.

Fruits and vegetables are frequently affected by pathogenic microorganisms. For example, gum arabic exhibited promising antimicrobial activity against pathogenic fungi, such as *C. gloeosporioides*, *Geotrichum citri-aurantii*, and *Penicillium digitatum*. Gum arabic and maize starch effectively coat pomegranate fruits for cold storage. In comparison with uncoated samples of pomegranate, the coating significantly reduced decay [91,92]. Additionally, other studies have demonstrated that gum arabic is effective at inhibiting the growth of fungi in perishable foods, such as avocado, mango, strawberries, and tomatoes, when used as packaging materials. 

Alginate-polysaccharides-based coatings minimize weight loss, preserve apple quality, and extend shelf life [93]. Gum-based edible coatings reduce respiration and oxidation rates by functioning as a semi-permeable barrier to oxygen and carbon dioxide [9]. López-Córdoba and Aldana-Usme [94] investigated the effect of coatings of sodium alginate and ascorbic acid on the physicochemical parameters of fresh-cut pineapple during storage. Their findings showed that alginate gum coatings could preserve the appearance of pineapple while refrigerated. According to Bal [95], edible coatings based on alginate gum influence preserving plum quality during post-harvest storage. The combination of alginate coating treatment and salicylic acid showed promising results in extending plum storage life for 40 days at 0–1 °C. Tomatoes can be preserved by polysaccharides coatings that act as vapor, solute, and gas barriers. Das et al. [96] studied alginate-based edible coatings on tomatoes to prevent postharvest ripening and retain quality. Their findings demonstrated the benefits of delaying ripening by lowering respiration rate, ethylene production, and ethylene-induced alterations, such as color change and loss of firmness. 

Chitosan coatings increase the storability of fresh produce by slowing respiration and reducing water loss. Chitosan coatings protect against bacterial contamination and moisture loss from food products’ surfaces, thereby extending their shelf life. A chitosan-based edible coating incorporating cellulose nanofibers and curcumin has been shown to prolong kiwi fruits’ storability and quality, including less weight and firmness loss, slow respiration rate, and a decrease in microbial growth [97]. Sun et al. [98] previously examined the shelf life of kiwis, strawberries, nectarines, avocados, apricots, and bananas coated with three bio-based nanomaterials of chitin, chitosan, and cellulose. The most potent antifungal agents were wood nanocrystals and chitosan nanofibers. Kumar et al. [28] evaluated green bell pepper samples’ shelf life and quality parameters using chitosan-pullulan and pomegranate peel extract. The coatings considerably reduced physiological weight loss and color browning. Moreover, the chitosan also inhibits the growth of fungi, which degrades the quality of fresh-cut cucumbers when they are stored [99]. In addition, the papaya, which experiences a 20–30% post-harvest loss, was also studied [100]. Post-harvest degradation of papayas makes them a target for infections, reducing their acceptability and shelf life.

Similarly, another study conveyed that the ripening of peanut samples was delayed by using a pullulan, pectin, and grape seed extract as bioactive binary coatings [101]. As a result, there was a decrease in lipid oxidation, which delayed their rancidity, thereby improving the storability of coated samples. Antibacterial activity of the pectin, pullulan, and grape seed film was demonstrated against *E. coli* and *L. monocytogenes*. Lara et al. [102] studied the effects of fresh-cut lotus roots coated with different xanthan gum concentrations. Citric acid was present in all previously mentioned solutions as an anti-browning agent, which reduced the enzymatic browning of fresh-cut lotus roots and glycerol was included as a plasticizer. The treated samples exhibited a lower microbial count than the untreated fresh-cut lotus root samples after 24 h. Khorram et al. [103] investigated the effects of different compounds, including edible gelatin, Persian gum, and shellac, as coatings for oranges. The results show that different coating agents enhanced shelf life and physical-chemical qualities. The most notable benefits for fresh-cut fruits are reduced water loss, enhanced solid soluble content, and color preservation.

**Table 1 polymers-14-01341-t001:** Examples of polysaccharide coatings employed to enhance the quality of various fruits and vegetables.

Polysaccharides	Additives/Surfactants	Coated Fruits/Vegetables	Effects on Fruits and Vegetables	References
Gum arabic	Glycerol	Ponkan orange (*Citrus poonensis*)	Reduced postharvest decay and membrane lipid peroxidation Maintained nutritional quality Retarded fruit quality deterioration	[91]
Gum arabic, maize starch	Lemongrass oil, glycerol	Pomegranate (*Punica granatum*)	Prevented weight loss during storage to maintain quality	[92]
Gum arabic	Glycerol	Strawberry (*Fragaria ananassa*)	Inhibited fungal growth completely Preserved visual quality	[73]
Gum arabic	-	Tomato (*Solanum lycopersicum*)	Reduced water activity during storage	[104]
Gum arabic, cellulose	Moringa leaf extract	Avocado (*Persea americana*)	Retarded weight loss Delayed color changes and inhibited growth of *C. gleosporioides*	[105]
Gum arabic	*Aloe vera* gel, ethanolic	Mango (*Mangifera indica*)	Prevented weight loss Reduced acidity loss Delayed ripening process	[106]
Alginate	Citric acid, ascorbic acid	Apple (*Malus pumila*)	Reduced weight loss and microbial contamination	[93]
Alginate	Ascorbic acid	Pineapple (*Ananas comosus*)	Preserved the color Inhibited the polyphenol oxidase	[94]
Alginate	Salicylic acid, oxalic acid	Plum (*Prunus salicina* L. cv. ‘Black amber’)	Reduced weight reduction Delayed in respiration rate changes Extended shelf life for 40 days	[95]
Alginate	Orange essential oil, Tween 80	Tomato	Inhibited the growth of bacteria Prevented ripening and spoilage	[96]
Chitosan, cellulose	Curcumin	Kiwi fruit (*Actinidia deliciosa*)	Minimized weight loss, firmness loss, respiration rate, and microbial count for 10 days storage at 10 °C	[97]
Chitosan, chitin, cellulose	-	Kiwi, avocado, strawberry, banana, nectarine, apricot	Retained fruit freshness with excellent antifungal action	[98]
Chitosan, pullulan	Pomegranate peel extract	Green bell pepper (*Capsicum annuum*)	Retained phenolic, flavonoid, and antioxidant properties Retained physicochemical properties Retained organoleptic quality over 18 days	[28]
Chitosan	Acetic acid	Cucumber (*Cucumis sativus*)	Preserved fresh-cut cucumber freshness and shelf life up to 12 days Reduced fungal count	[99]
Chitosan	Calcium chloride	Papaya (*Carica papaya* L.)	Extended storability Reduced the growth of decay-causing fungi	[100]
Pectin, pullulan	Grape seed extract (*Vitis vinifera*)	Peanut (*Arachis hypogaea*)	Reduced bacterial growth and rancidity Prevented lipid oxidation and prolonged shelf life	[101]
Xanthan gum	Citric acid, glycerol	Lotus root (*Nelumbo nucifera*)	Decreased enzymatic browning Inhibited growth of Bacillus subtilis	[102]
Persian gum	Gelatin, shellac	‘Valencia’ orange	Reduced water loss and imparted gloss Reduced weight and firmness loss	[103]

## 5. Preservation Mechanisms

Due to internal and external factors, the post-harvest quality of fruits and vegetables may deteriorate. Recent advances in polysaccharide-based edible coatings have demonstrated favorable effects on preserving fresh fruit quality, most notably by providing antibacterial, antifungal, and antioxidant qualities. As indicated in Table 2, these systems aim to use the polysaccharide’s barrier properties against physical and mechanical impacts, chemical reactions, and microbial invasion. Polysaccharides are also biodegradable, readily available, and inexpensive. This latter reduces the need for non-biodegradable synthetic packaging materials. Polysaccharides can also be modified to improve their physicochemical properties.

The polysaccharides-based coating provided antimicrobial preservative mechanisms on the fresh produce. Du et al. [107] stated that a combination of chitosan and sodium alginate (SA/CS) had shown antibacterial activity by preventing mold growth on strawberries. Chitosan polysaccharide enriched with cinnamon oil has decreased the rate of respiration and controlled the ripening of sweet cherries during storage by increasing the exosmosis ratios of Penicillium citrinum and Aspergillus flavus, where the development of spores and mycelium was inhibited and killed [108]. In addition, Nourozi and Sayyari [69] have proposed that aloe vera gel-based coverings coated with basil seed mucilage imparted antimicrobial preservative mechanisms by preventing pathogen growth. Moreover, cationic starch and anionic sodium alginate were reported to provide antimicrobial properties via the inhibition zone method [109]. Previously, Wu [110] said that polysaccharides of cactus *Opuntia dillenii* showed bacteriostatic activity, which can inhibit the growth of bacteria.

Besides providing antimicrobial properties to fresh produce, polysaccharides-based coatings also anticipated antifungal properties, which inhibited fungus that caused rot to appear on the crop. Previously, Nicolau-Lapena [111] has mentioned that aloe vera gel possessed antifungal preservative mechanisms where the bioactive components of aloe vera gel aided in enhancing the quality attributes of fruits, such as grapes, tomatoes, peach, sweet cherry, litchi fruit, fresh-cut papaya, white button mushroom, fresh-cut guava, and pomegranate arils. Similarly, Nair et al. [112] reported that a combination of chitosan and alginate enriched with pomegranate peel extract is indicated to inhibit *Colletotrichum gloeosporioides* by disrupting the cell wall of the fungal. The addition of nanoparticles, such as ZnO, in the antimicrobial coating of alginate and chitosan manifested extra antifungal activity by inhibiting the growth of fungus *Phyllosticta psidicola* [109]. Correspondingly, Kharchoufi et al. [113] has disclosed that the coating of chitosan and locust bean gum enriched with pomegranate peel extract significantly inhibited *P. digitatum* in oranges. Hajji et al. [114] also reported the findings of the antifungal activity found on strawberries coated with a combination of chitosan enriched with carotenoproteins coatings. When used alone, the enhancement of dietary fibers and essential oils onto chitosan coating generated superior antifungal properties than chitosan [115]. 

Polysaccharides-based coatings have been reported to improve the antioxidant properties of coated fruits and vegetables to improve the product’s titratable acid, vitamin C, and phenolic content. Aloe vera gel delivers antimicrobial properties in a fruit coating and provides antioxidant properties to the fresh produce due to bioactive components in aloe vera gel [111]. Earlier, Alvarez et al. [115] reported incorporating dietary fibers, such as oligofructose and orange fiber, into sodium alginate, and a chitosan coating provides antioxidant properties due to high content phytochemicals and phenols in it. Notably, chitosan alone retains antioxidant properties, thus inhibiting browning and controlling the overproduction of ROS [116]. Panahirad et al. [117] discovered that carboxymethylcellulose coating assists the preservation of plum fruit firmness, titratable acidity, vitamin C, anthocyanin, and flavonoid content to its antioxidant qualities. 

Previously, researchers reported that the effectiveness of polysaccharide-based coatings in inhibiting respiration rate in fruits might be linked to engendering properties of polysaccharide-based coatings, such as providing enzyme-related defense mechanisms [116,118] and the ability of the coating to partially fill the apertures of the fruits’ dermal tissue, resulting in a significant reduction of gas exchange and respiration [119]. The polysaccharides-based coating may act as a semi-permeable barrier and serve as modified atmospheric packaging to shield fruits from oxygen, carbon dioxide, and moisture, hence lowering the rate of fresh product respiration [120].

**Table 2 polymers-14-01341-t002:** Application of polysaccharide’s barrier properties against physical and mechanical impacts, chemical reactions, and microbial invasion.

Polysaccharides Used	Fruits/Vegetables Used	Target	Preservative Mechanisms	Effect on Fresh Produce	References
Sodium alginate + Chitosan (SA/CS)_3_	Strawberry (*Fragaria ananassa*)	Microbial cell membrane	Antimicrobial mechanisms of chitosan: Positively charged chitosan molecules interacting with negatively charged bacteria membranes inhibit microbial growth and toxin buildup.	Provided antibacterial properties Restricted gas exchange and water loss Provided excellent oxygen and water vapor barrier characteristics Reduced weight loss Reduced oxidative damage during storage Inhibited mold growth	[107]
Chitosan(CH) + cinnamon oil	Sweet cherry (*Prunus avium* L.)	Cell wall and cell membrane	Antimicrobial mechanisms of chitosan: The presence of chitosan micropores as a gas barrier and a carrier for cinnamon oil, along with: The presence of hydroxyl groups inhibits mycotoxin formation by creating hydrogen bonds with active enzymes. Antimicrobial mechanisms of cinnamon oil: Trans-cinnamaldehyde, a component in cinnamon, was shown to induce unevenness of hyphae cell walls of fungus, mechanical damage, and disrupted cell metabolism.	Decreased rate of respiration Decreased O_2_ level Increased CO_2_ in cherry fruit packaging Controlled the decay of fruits during storage	[108]
Aloe vera gel + Basil seed mucilage (AVG + BSM)	Apricot (*Prunus armeanica cv. ‘Nouri’*)	-	Antimicrobial mechanisms of AVG and BSM: TPC accumulated in response to AVG and BSM. Phenolic chemicals directly affect the defense process by inhibiting pathogen growth and reinforcing host tissues.	Reduced weight loss Decreased soluble solid Slowed respiration rate Lowered ethylene output Lowered ripening index Maintained fitness of fruits	[69]
Aloe vera gel (AVG)	Grapes, Tomatoes, Peach, Sweet cherry, Litchi fruit, fresh-cut papaya, white button mushroom, fresh-cut guava, pomegranate arils	Phospholipid bilayer of fungal	Antifungal mechanisms of AVG: Bioactive compounds of aloe vera gel, Aloin, and Barbaloin disrupt the lipid/water interface in negatively charged phospholipids, causing fungal bilayer core rupture. Antioxidant mechanisms of AVG: Aloe-emodin, one of the bioactive components in AVG, prevents the degradation of flavonoid, FC and total phenolic content, TPC.	Decreased respiration Decreased ripening process Prevented browning reaction Delayed firmness loss Inhibited weight loss Retained phenolic content	[111]
Chitosan (CH) + Alginate (AL) + Pomegranate peel extract (PPE)	Capsicum (*Capsicum annuum* L.)	Cell wall of the microorganisms	Antifungal mechanisms of pomegranate peel extract (PPE): PPE is high in phenolics, which are antioxidants and antimicrobial compounds.	Inhibited Colletotrichum gloeosporioides Retained weight, firmness, color, and ascorbic acid Inhibited the microbial growth Maintained organoleptic properties Extended the storability up to 25 days at 10 °C	[112]
Guar GUM (GG) + Ginseng extract (GSE)	Sweet cherry (*Prunus avium* L.)	-	Preservative mechanisms of GG-GSE: The formation of an extended network between GG and hydrophilic molecules, such as phenols via hydrogen bonding reduces respiration rate, water loss, and oxidation reaction. Enzyme related-defense mechanisms: High GSE bioactive component concentration reduces the oxygen supply needed for enzymatic oxidation of phenols.	Controlled water loss Retained firmness, ascorbic acid, and total phenols Increased sweet cherry shelf life by 8 days	[118]
Alginate + Chitosan + ZnO nanoparticles	Guava (*Psidium guajava* L.)	-	Antifungal activity of nanoZnO: ZnO nanoparticles prevent the development of *Phyllosticta psidicola* fungus when coated with alginate and chitosan. Antimicrobial activity of cationic starch and sodium alginate: Cationic groups are acquired via the production of cationic starch. Anionic groups were created using sodium alginate, a natural anionic polysaccharide. The antibacterial activity is measured by zone inhibition (disc diffusion method).	Prevented rot appearance of fruits, especially the fungus Phyllosticta psidicola Prevented bulk loss in fruits Retarded maturational physicochemical changes Preserved guava for 20 days vs. 7 days for uncoated fruits	[109]
Citrus pectin + broken rice grain flour + cellulosic rice skin nanofiber	Avocado (*Persea americana Milll.*)	Apertures present in the dermal tissue of the fruit	Preservative mechanism of reducing respiratory rate: Coating essentially seals the dermal tissue’s apertures, severely reducing gas exchange or respiration. Coatings partially block natural fruit apertures, acting as a modified atmosphere package, slowing metabolism and enzyme activity, notably pectinases.	Preserved avocado green color Reduced respiration rate Provided firm appearance on avocado Extended shelf life Delayed ripening by a minimum of 8 days	[119]
Chitosan (CH) + Locust Bean Gum (LBG) + Pomegranate Peel Extract (PPE)	Orange (*Citrus sinensis*)	-	Antifungal mechanisms of the coating: Inhibition halo requires both CH and LBG. CH has an inherent inhibitory action. The *P. digitatum* growth was considerably reduced by CH and LBG coatings supplemented with PPE.	Reduced disease incidence significantly. Controlled Penicillium digitatum growth Reduced postharvest decay	[113]
Chitosan + carotenoproteins	Strawberries (*Fragaria ananassa*)	Fungal reproductive structure	Antifungal activity of the coating: For maximum bacterial suppression, use higher CS and CP concentrations in edible film creation. CS and CP improved fungicidal growth inhibition. CS’s potency was related to its ability to alter the morphology of fungal reproductive structures.	Reduced the fungal decay Reduced weight loss Inhibited phytopathogenic growth on cold-stored fruit Extended shelf-life of strawberries	[114]
Sodium Alginate + Chitosan + Different Dietary Fibers (apple fiber, orange fiber, inulin, oligofructose)	Blueberries (*Vaccinium sect. Cyanococcus*)	Cell membrane	Antioxidant activity of oligofructose and orange fiber: The high amount of phytochemicals and phenols as well as an antioxidant capacity that triggers the phenylpropanoid metabolism allowed more significant reductions in yeast/mold counts than CH alone. Antifungal activity of CH–OL and CH–OF: OF and OL extract may include plant-derived bioactive substances with antifungal activity.	Inhibited the growth of mesophilic bacteria and yeasts/molds Reduced degradation rate by greater than 50% Boosted antioxidant properties Retained fruit firmness Halted off-odor development Enhanced visual quality Increased blueberry sensory shelf life by 6 days	[115]
Chitosan + Lemongrass oil	Bell pepper (*C. capsici*)	Cell wall and cell membrane	Antifungal mechanism of CH and EO: The capacity of EO to permeate cell membranes damages biological processes, such as ion loss, coagulation of bacterial cytoplasm, and direct damage to proton pump and ATP generation sites. A reduction in membrane potential causes calcium ion leakage and other vital components.	Controlled fungal growth of anthracnose Maintained food quality	[121]
Carboxymethylcellulose (CMC)	Plum fruits (*Prunus domestica* L.)	-	Antioxidant mechanism of CMC: The presence of carboxylic groups in CMC’s chemical structure results in hydrogen bonding within the coating matrix and between the coating and the fruit peel. Limiting O_2_ access and changing internal gas composition, lowering oxidative metabolism via increased peroxidase activity, and delaying textural changes in fruits.	Preserved firmness, anthocyanin, flavonoid, titratable acidity, vitamin C, anthocyanin, and flavonoid content Maintained antioxidant capacity of plum fruits Increased peroxidase activity Decreased polyphenol oxidase and polygalacturonase	[117]
Cactus *Opuntia dillenii* polysaccharide (ODP)	Fresh-cut potato	Bacterial cellular activity	Antimicrobial mechanism of ODP: ODP’s bacteriostatic effect slows bacterial growth by interfering with bacterial protein production, DNA replication, or other cellular metabolisms.	Suppressed browning Decreased microbial growth Reduced respiration rate Inhibited weight loss Inhibited total sugar formation	[110]
Fenugreek and flax polysaccharides coating	Apples (*Malus domestica*)	-	Preservative mechanism of reducing respiratory rate: Coating material alters the environment of fruit by forming a thick surface layer, which reduces the pace of respiration and the degradation process by preventing gas exchange.	Prevented weight loss Improved firmness of fruits Decreased respiration rate Slowed down the increment in TSS of apple during storage Slowed down the pH changes in apples Maintained TA of coated apple	[122]
Alginate + Chitosan + Carrageenan	Fresh-cut lettuce (*Lactuca sativa* L. *var. ramosa Hort*.)	-	Antioxidant mechanisms of the coating: Chitosan inhibits browning and increases antioxidant properties and controls the overproduction of ROS, and suppresses lipid peroxidation. Enzyme related-defense mechanisms: Inhibit the polyphenol oxidase (PPO) activity and postpone the time to reach its maximum level while triggering the formation of phenylalanine ammonia-lyase (PAL). Inhibit the phospholipase D, PLD and lipoxygenase, LOX activity while dramatically maintaining high activities of antioxidant enzyme (catalase, CAT; peroxidase, POD; superoxide dismutase, SOD; and ascorbate peroxidase, APX.	Inhibited enzymatic browning Lowered malondialdehyde content Enhanced antioxidant enzyme Postponed senescence	[116]
Polysaccharide isolated from *Oudemansiella radicata*	Shiitake mushrooms (*Lentinus edodes*)	-	Preservative mechanism of reducing respiratory rate: Capability to operate as a semi-permeable barrier to the flow of O_2_, CO_2_, moisture, and solutes, as well as to generate a modified environment to limit respiration rate. Coatings delay the loss of natural volatile components, allowing the original mushroom flavor to be preserved for a longer shelf life.	Reduced weight loss Enhanced firmness Decreased browning Lowered malondialdehyde content Increased physical microstructure Lowered enzyme activities: protease, polyphenol oxidase, peroxidase, phenylalanine ammonia lyase, cellulase, chitinase Produced mushrooms with high concentrations of superoxide dismutase and catalase, monosodium glutamate-resembling amino acids, umami 5′-nucleotides, and 1-octen-3-ol	[120]

## 6. Legislations and Safety Issues of Edible Films and Coatings

Polysaccharide coatings are regulated similarly to other dietary components and act as an essential component of fruits and vegetables [123]. It can be used in developing edible packaging if a substance has been approved as safe by the FDA, is categorized as GRAS, and is used by Good Manufacturing Practices (GMP). If the biopolymer component utilized is still not GRAS-approved, the producer who wants to use it may request it, assuming its safety can be demonstrated. Besides, the producer can submit a GRAS Affirmation Petition to FDA or commercialize the product without FDA approval. Furthermore, active and intelligent packaging might accommodate antimicrobials, antioxidants, colors, and other nutrients to improve the function of the films or coatings. EU regulations required food additives to be identified on packaging with their functional category and either their name or E-number [124]. 

Regulatory statutes cover other significant issues related to allergenic components in edible films and coatings. Protein residues are common in polysaccharide extracts [125]. For instance, guar gum extract might contain less than 10% protein, which may trigger allergies [126]. In some cases, occupational asthma and allergies have been linked to guar gum or guar gum dust exposure [127]. Nonetheless, guar gum is not a significant food allergy due to the small number of cases reported [128]. Aside from polysaccharides, edible films and coatings can be manufactured using wheat (gluten), milk (whey, casein), soy, and peanut proteins, which are prominent as allergens [129]. 

Most countries consider chemical compounds introduced as antimicrobials as food additives if their primary intent is to lengthen the shelf life of fruits and vegetables. National regulations govern the application of edible coatings and their concentrations. The coating is used in nations where the fresh product is exported. Generally, all substances listed on the label must be accurately described because the edible films and coatings have become an inseparable part of fresh food [124]. 

## 7. Benefits and Limitations of Natural Packaging

Natural-based films provide several advantages over directly adding preservatives to food items to increase their shelf life, quality, and safety. By embedding preservatives into the packing material, just the appropriate quantity of antibacterial or antioxidant agents is employed, minimizing preservative interaction with the food product. Furthermore, including antimicrobial and antioxidant compounds into films keeps them from seeping across the food matrix and reacting with other food components, such as proteins and lipids, resulting in a loss of effect. Another benefit of antimicrobial/antioxidant film is that it allows the release of these agents in a controlled way. Figure 3 depicts a review of the advantages and disadvantages of natural packaging.

## 8. Conclusions and Future Recommendations

According to the extensive analysis included in this review, it was concluded that polysaccharides-based edible packaging is an environmentally friendly technology that utilizes biodegradable ingredient substances to effectively prolong shelf life of fresh produce. The FDA and other food safety regulatory agencies have approved the edible coating restrictions and recognized polysaccharides as GRAS. Edible coating and film packaging technology has been a boon for agricultural and horticultural products. Understanding how polysaccharides-based edible coatings and film forming solutions contribute to the preservation of food products’ physicochemical qualities and shelf life, as well as to the development of novel formulas for prepared coating solutions, is important. The edible coating and film are both gas and water resistant. Researchers have concentrated their efforts over the last few years on producing highly functional, nanostructured, and multilayered polysaccharides-based edible coating and film materials in a variety of combinations and concentrations. The commodity-specific application technology would be used. Additionally, this review study revealed previously created edible films enriched with other substances, as well as their influence on fruits and vegetables. Further research should focus on the use and technique for commercializing highly functional polysaccharides-based coating and film forming materials. 

## Figures and Tables

**Figure 1 polymers-14-01341-f001:**
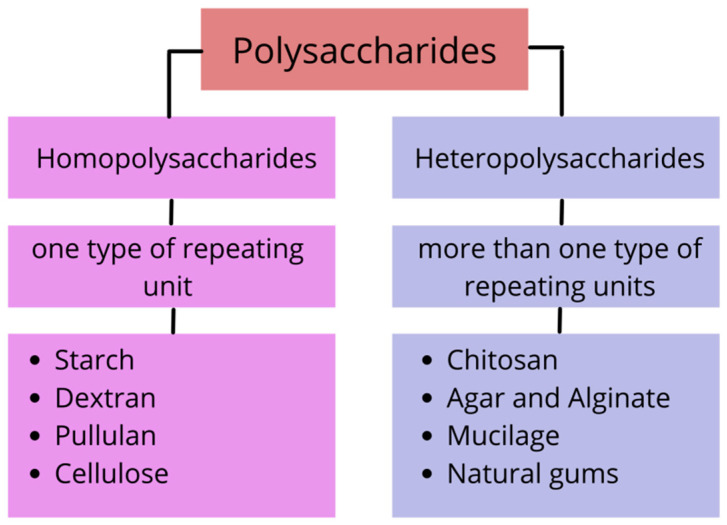
Classification of polysaccharides as an edible coating.

**Figure 2 polymers-14-01341-f002:**
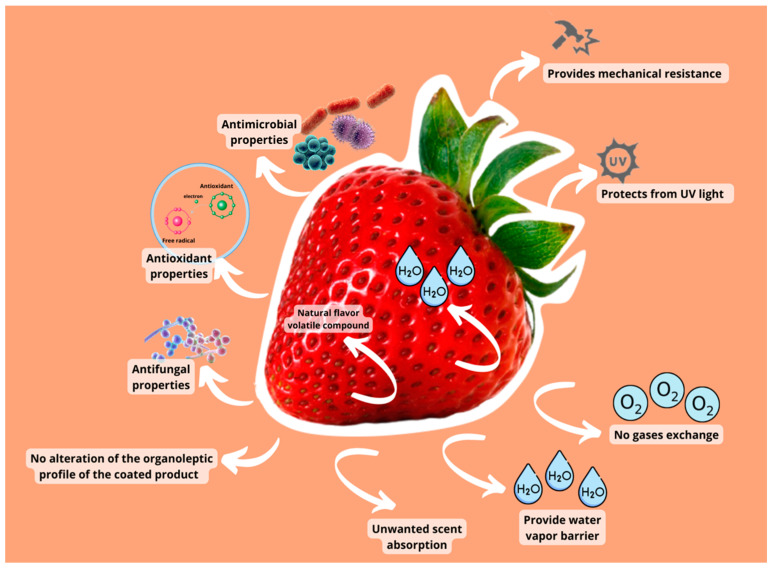
The direct correlations between coating qualities and the quality variables preserved in fruits and vegetables.

**Figure 3 polymers-14-01341-f003:**
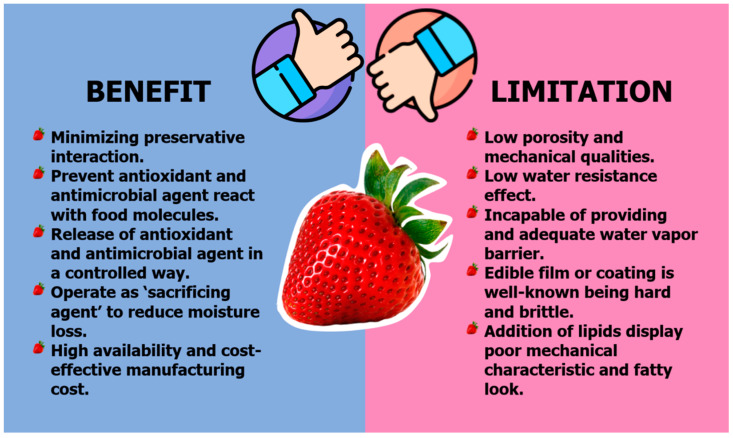
Summary of benefits and limitations of natural packaging.

## Data Availability

Not applicable.

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
