# Peer review of "Recent Advancements of Polysaccharides to Enhance Quality and Delay Ripening of Fresh Produce: A Review"

_polymers, 2022, doi:10.3390/polym14071341_

Round 1
Reviewer 1 Report
It is a interesting a review article describing the different types of
polysaccharides and their novel applications as natural food preservatives.
The work is basically well done and properly described but I’m not convinced that it has enough citations. The authors should be supplemented by literature data for more recent positions.
It will be interestingly to add a few references about modify the biodegradable polymers with the natural vegetable compounds example carvacrol, tymol, eugenol... and maby to add in the introduction about physical modify packaging materials.
Author Response
Dear Editor,
The authors would like to inform you that the revised manuscript referred to above has corrected the suggestions and constructive comments of the reviewers and editors. The comments and suggestions highlighted by the reviewers were considered in the revised manuscript. The modifications, additions, and corrections appear in the highlight within the article. The following point to point is given as follows:
Reviewer #1:
It is a interesting a review article describing the different types of polysaccharides and their novel applications as natural food preservatives. The work is basically well done and properly described but I’m not convinced that it has enough citations. The authors should be supplemented by literature data for more recent positions. It will be interestingly to add a few references about modify the biodegradable polymers with the natural vegetable compounds example carvacrol, tymol, eugenol... and maby to add in the introduction about physical modify packaging materials.
- Thank you for the suggestion. The manuscript has been checked and revised. The authors have added a few more references regarding modified biodegradable polymers with the natural vegetable compounds as suggested in Section 2.
- The suggestion regarding adding physical modified packaging material is accepted. Hence, a new paragraph has been added in Section 1 (Introduction):
“Nonetheless, the surface polymers’ particular nature properties precludes their vast range of intended applications, necessitating their modification via a variety of techniques. For instance, the presence of hydroxyl groups renders the surfaces hydrophilic, a feature that is not inherent in other polymeric surfaces. Thus, it is critical that intrinsically nonconforming surfaces undergo treatment in order to ease their usage in a variety of industrial applications and overcome their significant limitations. Polymer modification is a frequently utilised technique for imposing desired short and long-terms effects necessary for their efficient performance. While surface treatments improve the performance of current materials, they also expand the application range of polymers due to its versatility of bulk material qualities such as elasticity, tensile strength, and density (Nemani et al. 2018).”
Reviewer 2 Report
This manuscript provides the progress in polysaccharide based coating for fresh produce. Some part of the manuscript seems to be textbook rather than review article which shows the advance data. However, most parts are importance and give up-to-date data which can be a good reference. Here are points to be revised.
Abstract
L12 What does “latest trend catering to society’s” mean?
Introduction
L58 The word “involved” should be revised.
The aim/objective of the research should be clearly stated in the last paragraph
Section 2 provides data that seems to be available in most of polysaccharide textbook. There should be more recent investigation/discussion which shows advancement.
L210 Why tensile strength (stretching) is necessary for the coating?
Table 1 The writing/phrase should be the same. For example, most of the sentence in the forth column start with Verb+ed.
[75] remove “It”
[82] [85] replace 1. With bullet
L332-333 antifungal is one kind of antimicrobial. Please revise the sentence
Table 2 Recheck the line and alphabet to be the same row e.g. [96] “Phospholipid”, [99] “reproductive”, [103] “polygalacturonase”
[96] It should be “degradation of flavonoid” remove “content”.
[97] Remove “The”
Recheck use of number and bullets to be the same.
L375-377 Rewrite the sentence
L413 Antimicrobial packaging can be an alternative to direct addition of preservative into food products (Leelaphiwat, 2022 Food Chemistry).
L414 Previous investigations developed active films to extend shelf-life of meat (Chatkitanan, 2020 Food Packaging and Shelf-life), seafood (Laorenza 2021 Food Chemistry), bakery (Klinmalai, 2021) and fresh produce (Phothisarattana, 2021).
L423 Blending of biopolymers was efficient to control and achieve desirable mechanical and barrier properties by modified morphology of polysaccharides (https://doi.org/10.1002/app.45533).
Conclusions
L454-457 does not relate to polysaccharides. The recommendation should be revised to emphasize/strengthen the knowledge of polysaccharide/coating.
Author Response
Dear Reviewer,
The authors would like to inform you that the revised manuscript referred to above has corrected the suggestions and constructive comments of the reviewers and editors. The comments and suggestions highlighted by the reviewers were considered in the revised manuscript. The modifications, additions, and corrections appear in the highlight within the article. The following point to point is given as follows:
This manuscript provides the progress in polysaccharide based coating for fresh produce. Some part of the manuscript seems to be textbook rather than review article which shows the advance data. However, most parts are importance and give up-to-date data which can be a good reference. Here are points to be revised.
Abstract
- L12 What does “latest trend catering to society’s” mean?
- The suggestion has been revised to “Fresh-cut products are the latest trend in fulfilling society's restless needs, and the food industry is faced with the challenge of maintaining the quality of fresh produce.”
Introduction
- L58 The word “involved” should be revised.
- The word has been revised to “applied on”
- The aim/objective of the research should be clearly stated in the last paragraph
- The aim/objective of the review has been stated in the last paragraph in the Section 1.
“This review informs readers about the different types of polysaccharides and their novel applications as natural food preservatives.”
- Section 2 provides data that seems to be available in most of polysaccharide textbook. There should be more recent investigation/discussion which shows advancement.
- Thank you for the comments and suggestions. The authors have included more recent advancements regarding polysaccharides in Section 2.
- L210 Why tensile strength (stretching) is necessary for the coating?
- Tensile strength (stretching) is necessary for the coating as can be seen in the statement of “Tensile strength and stretchability are crucial film parameters that can be utilised to monitor the film’s integrity and resistance to environmental stress throughout the coating treatments.”
- Table 1 The writing/phrase should be the same. For example, most of the sentence in the forth column start with Verb+ed.
- The writing/phrase of the fourth column has been revised. All the sentences have been revised into Verb+ed writing style.
- [75] remove “It”
- The word “It” has been removed as suggested.
- [82] [85] replace 1. With bullet
- The sentences have been divided into bullet points.
[82]
- Inhibited the growth of bacteria
- Prevented ripening and spoilage
[85]
- Retained phenolic, flavonoid, and antioxidant properties
- Retained physicochemical properties
- Retained organoleptic quality over 18 days
- L332-333 antifungal is one kind of antimicrobial. Please revise the sentence
- The sentences have been revised accordingly
- Table 2 Recheck the line and alphabet to be the same row e.g. [96] “Phospholipid”, [99] “reproductive”, [103] “polygalacturonase”
- The line and alphabets in Table 2 have been re-checked.
- [96] It should be “degradation of flavonoid” remove “content”.
- The word “content” has been removed as suggested.
- [97] Remove “The”
- The word “The” has been removed as suggested.
- Recheck use of number and bullets to be the same.
- The use of numbers and bullets have been re-checked.
- L375-377 Rewrite the sentence
- The sentences have been revised accordingly
- L413 Antimicrobial packaging can be an alternative to direct addition of preservative into food products (Leelaphiwat, 2022 Food Chemistry). L414 Previous investigations developed active films to extend shelf-life of meat (Chatkitanan, 2020 Food Packaging and Shelf-life), seafood (Laorenza 2021 Food Chemistry), bakery (Klinmalai, 2021) and fresh produce (Phothisarattana, 2021). L423 Blending of biopolymers was efficient to control and achieve desirable mechanical and barrier properties by modified morphology of polysaccharides (https://doi.org/10.1002/app.45533).
- The important information has been included in the manuscript.
- https://www.sciencedirect.com/science/article/abs/pii/S0308814621019622?casa_token=iMvjWL5rBtkAAAAA:MC72YG8SOV3abkJeQaJse27yQHTCXCY3OZel3uhRyz3oBZeBVRXbLruzYcdrcnlCTTrJQarKJ6Z0
- https://www.sciencedirect.com/science/article/abs/pii/S2214289420300831?casa_token=HRG_H0sRx7cAAAAA:SUGrE_vYxHBwULyB56X_Mcnrir42ZlrErtjCCWkLUq7zv1ndVU-KrmRyCD4GNVzSqaTsZ9r7wbts (Chatkitanan, 2020)
- https://www.sciencedirect.com/science/article/pii/S0023643821015097?casa_token=r9EdkOSFNUEAAAAA:ZVgTeAemdV83GacCcpNyzCoA1PuenQo52WnRi4aEkLchy0YdzG7iYbhdHVlU3_cOO4sGLC_3lhbL(Klinmalai, 2021)
- https://www.mdpi.com/2073-4360/13/23/4192 (Phothisarattana, 2021)
- https://www.sciencedirect.com/science/article/abs/pii/S0308814621012589 (Laorenza, 2021)
- https://onlinelibrary.wiley.com/doi/abs/10.1002/app.45533
Conclusions
- L454-457 does not relate to polysaccharides. The recommendation should be revised to emphasize/strengthen the knowledge of polysaccharide/coating.
- The conclusion and recommendation has been revised as suggested:
“According to the extensive analysis included in this review, it was concluded that polysaccharide-based edible packaging is an environmentally friendly technology that utilises biodegradable ingredients substances to effectively prolong shelf-life of the fresh produce. The FDA and other food safety regulatory agencies have approved the edible coating restrictions and recognized polysaccharides as GRAS. Edible coating and film packaging technology has been a boon for agricultural and horticultural products. Understand how polysaccharide-based edible coatings and film forming solutions contribute to the preservation of food products' physicochemical qualities and shelf life, as well as to the development of novel formulas for prepared coating solutions. The edible coating and film are both gas and water resistant. Researchers have concentrated their efforts over the last few years on producing highly functional, nanostructured, and multilayered polysaccharide-based edible coating and film materials in a variety of combinations and concentrations. The commodity-specific application technology would be used. Additionally, this review study revealed previously created edible films enriched with other substances, as well as their influence on fruits and vegetables. Further research should focus on the use and technique for commercialising highly functional polysaccharide-based coating and film forming materials.”
Round 2
Reviewer 2 Report
The authors have not yet revised the manuscript as recommend. Please also carefully recheck the citation in revise version to comply with the response to reviewer.
Section : Toxicity and allergenicity of polysaccharide coatings
Discussion in this part is still unclear. This part also make the reader confused about the safety of polysaccharide coating.
Author Response
Manuscript ID: membranes-1519181
Manuscript Title: Recent advancements of polysaccharides to enhance the quality and delay the ripening of fresh produce: A Review
Dear Reviewer,
The authors would like to inform you that the revised manuscript referred to above has corrected the suggestions and constructive comments of the reviewers and editors. The comments and suggestions highlighted by the reviewers were considered in the revised manuscript. The modifications, additions, and corrections appear in the highlight within the article. The following point to point is given as follows:
Reviewer #2:
The authors have not yet revised the manuscript as recommended. Please also carefully recheck the citation in the revised version to comply with the response to the reviewer.
- Thank you for the comments and recommendations. The revised citations have been carefully rechecked.
- The authors have revised the manuscript as recommended. The authors successfully cited two related research papers as per recommended. Only two research papers have been included in the manuscript as this manuscript focuses on preservation of fruits and vegetables.
- Structure–property modification of microcrystalline cellulose film using agar and propylene glycol alginate (Reference [55])
- Biodegradable Poly (Butylene Adipate-Co-Terephthalate) and Thermoplastic Starch-Blended TiO2 Nanocomposite Blown Films as Functional Active Packaging of Fresh Fruit (Reference [82])
Section: Toxicity and allergenicity of polysaccharide coatings. Discussion in this part is still unclear. This part also makes the reader confused about the safety of polysaccharide coating.
- Thank you for the comment. Section “Toxicity and allergenicity of polysaccharide coatings” have been revised to “ Legislations and Safety Issues of Edible Films and Coatings.” Besides, the discussion section of the particular topic have been revised as follow:
Polysaccharide coatings are regulated similarly to other dietary components and act as an essential component of fruits and vegetables [125]. It can be used in developing edible packaging if a substance has been approved as safe by the FDA, is categorized as GRAS, and is used by Good Manufacturing Practices (GMP). If the biopolymer component utilized is still not GRAS-approved, the producer who wants to use it may request it, assuming its safety can be demonstrated. Besides, the producer can submit a GRAS Affirmation Petition to FDA or commercialize the product without FDA approval. Furthermore, active and intelligent packaging might accommodate antimicrobials, antioxidants, colors, and other nutrients to improve the function of the films or coatings. EU regulations required food additives to be identified on packaging with their functional category and either their name or E-number [126].
Regulatory statutes cover other significant issues related to allergenic components in edible films and coatings. Protein residues are common in polysaccharide extracts [127]. For instance, guar gum extract might contain less than 10% proteins which may trigger allergies [128]. In some cases, occupational asthma and allergies have been linked to guar gum or guar gum dust exposure [129]. Nonetheless, guar gum is not a significant food allergy due to the small number of cases reported [130]. Aside from polysaccharides, edible films and coatings can be manufactured using wheat (gluten), milk (whey, casein), soy and peanut proteins, which are prominent as allergens [131]. Therefore, toxicity and allergenicity of coatings should also be taken into account and must be explicitly notified to the consumer, with the relevant allergenic component warning, regardless of how little is applied during manufacturing. This is certainly relevant for fabricating essential oils as an antibacterial ingredient in edible coatings; although categorized and licensed by the European Commission and GRAS, these might as well have allergic effects. Consumption of larger dosages of these natural substances may cause harm [132]. Therefore, it is crucial to bridge the gap between the therapeutic efficacy of essential oils or plant extracts and the danger of toxicity [133].
Most countries consider chemical compounds introduced as antimicrobials as food additives if their primary intent is to lengthen the shelf life of fruits and vegetables. National regulations govern the application of edible coatings and their concentrations. The coating is used in nations where the fresh product is exported. Generally, all substances listed on the label must be accurately described because the edible films and coatings have become an inseparable part of fresh food [134]. Fabrication of antioxidants, antimicrobials, colorants, essential oils and other additives are subjected to the same regulations as food formulations. All materials and compounds in active packaging designed for interaction with food must be safe, according to Article 3 of the European Parliament and Council (EC) No. 1935/2004 and its amendments, as well as by Regulation No. 2023/2006 [135,136]. Active packaging is compulsory to have inert properties to prevent their constituents from migrating into food in proportions that endanger human health or produce unwanted changes in food attributes or organoleptic qualities under normal and unavoidable use situations [136,137].
Green packaging solutions, including the US Environmental Protection Agency's (EPA), recommended reducing initial packaging and encouraging the innovation of edible films constructed from pectin and other natural polymers. The development of a packaging system would help facilitate environmentally hazardous materials in packaging while also making it simple to recycle or compost. Furthermore, the FDA advised using packaging that reduces food damage or degradation, boosting the freshness and quality of packed meals [138]. By and large, edible coatings are harmless and biodegradable, with minimal toxicity to living things. Nevertheless, previous research has claimed that carrageenan gum has a detrimental impact. These deleterious effects are caused by degraded carrageenan (poligeenan), with a molecular weight of less than 50 kDa [139]. According to research made by Liu et al. [140], the carcinogenic effects of undegraded carrageenans were observed. When food-grade components are utilized in edible coatings and films, these detrimental effects of carrageenans are not seen [141].
Round 3
Reviewer 2 Report
“toxicity” should be carefully mentioned. The toxicology test is commonly performed with a new product or a new substance. The authors mentioned that “toxicity and allergenicity of coatings should also be taken into account and must be explicitly notified to the consumer”. The allergens shall be placed on the package to notify the consumers according to global legislation. However, the “toxicity” may not be placed on the package. This can cause misleading to the reader. If there is any toxic or harmful substances, it cannot be used as food.
The authors mentioned that essential oils (EO) are GRAS but “Consumption of larger dosages of these natural substances may cause harm [130].”. This can cause misleading as the GRAS of the EO mentioned about the intend to use for food. Moreover, the reader cannot find any statement about the harmful of EO which intended to be used as food in the literature [130] as the citation by authors. In the [130] there is only statement about the harmful as “Some EOs may cause allergic contact dermatitis in people who use them frequently, as occurs in aromatherapy (Bleasel et al., 2002, Trattner et al., 2008). Therefore, it is crucial to bridge the gap between the therapeutic efficacy of essential oils or plant extracts and the danger of toxicity [131]”. They are not the use in food. Mostly, the EO cannot be used in high amount in food applications as they impart the sensory characteristics. This citation may not relevance to the case of edible coating in fruits. Moreover, the [131] is the reference about the pharmaceutical applications which is out of scope for the edible coating in fruits.
The authors mention about the European Parliament and Council (EC) No. 1935/2004 which refers to the safety of active packaging. However, this regulation does not cover “coating materials forming part of the food and possibly being consumed with it” and “this Regulation should apply to covering or coating materials which cover cheese rinds, prepared meat products or fruit but which do not form part of food and are not intended to be consumed together with such food.”. The edible coating of fruits may not fall within the scope of this regulation. The discussion can cause misleading. Moreover, the reader doesn’t understand why the authors mention about the (EC) No 2023/2006 here. How is it relevance to the discussion on edible coating or polysaccharide applications in fruit?
The latter paragraph and discussion on “Green packaging…”, how are they relevance to the polysaccharide application in fruit or edible coating? The discussion is still unclear.
The statement is unclear and should be revised. “help facilitate environmentally hazardous materials in packaging while also making it simple to recycle or compost.”
Author Response
Manuscript ID: membranes-1519181
Manuscript Title: Recent advancements of polysaccharides to enhance the quality and delay the ripening of fresh produce: A Review
Dear Reviewer,
The authors would like to inform you that the revised manuscript referred to above has corrected the suggestions and constructive comments of the reviewers and editors. The comments and suggestions highlighted by the reviewers were considered in the revised manuscript. The modifications, additions, and corrections appear in the highlight within the article. The following point to point is given as follows:
Reviewer #2:
- “toxicity” should be carefully mentioned. The toxicology test is commonly performed with a new product or a new substance. The authors mentioned that “toxicity and allergenicity of coatings should also be taken into account and must be explicitly notified to the consumer”. The allergens shall be placed on the package to notify the consumers according to global legislation. However, the “toxicity” may not be placed on the package. This can cause misleading to the reader. If there is any toxic or harmful substances, it cannot be used as food.
- The information has been updated accordingly. The toxicity information has been deleted in the manuscript.
- The authors mentioned that essential oils (EO) are GRAS but “Consumption of larger dosages of these natural substances may cause harm [130].”. This can cause misleading as the GRAS of the EO mentioned about the intend to use for food. Moreover, the reader cannot find any statement about the harmful of EO which intended to be used as food in the literature [130] as the citation by authors. In the [130] there is only statement about the harmful as “Some EOs may cause allergic contact dermatitis in people who use them frequently, as occurs in aromatherapy (Bleasel et al., 2002, Trattner et al., 2008). Therefore, it is crucial to bridge the gap between the therapeutic efficacy of essential oils or plant extracts and the danger of toxicity [131]”. They are not the use in food. Mostly, the EO cannot be used in high amount in food applications as they impart the sensory characteristics. This citation may not relevance to the case of edible coating in fruits. Moreover, the [131] is the reference about the pharmaceutical applications which is out of scope for the edible coating in fruits.
- The information for references 130 and 131 has been deleted.
- The authors mention about the European Parliament and Council (EC) No. 1935/2004 which refers to the safety of active packaging. However, this regulation does not cover “coating materials forming part of the food and possibly being consumed with it” and “this Regulation should apply to covering or coating materials which cover cheese rinds, prepared meat products or fruit but which do not form part of food and are not intended to be consumed together with such food.”. The edible coating of fruits may not fall within the scope of this regulation. The discussion can cause misleading. Moreover, the reader doesn’t understand why the authors mention about the (EC) No 2023/2006 here. How is it relevance to the discussion on edible coating or polysaccharide applications in fruit? The latter paragraph and discussion on “Green packaging…”, how are they relevance to the polysaccharide application in fruit or edible coating? The discussion is still unclear. The statement is unclear and should be revised. “help facilitate environmentally hazardous materials in packaging while also making it simple to recycle or compost.”
- The information has been updated and deleted.